# Direct Oral Anticoagulants Are Associated with Superior Survival Outcomes than Warfarin in Patients with Head and Neck Cancers

**DOI:** 10.3390/cancers14030703

**Published:** 2022-01-29

**Authors:** Chien-Lin Lee, Wei-Shan Chen, Yinshen Wee, Ching-Shuen Wang, Wei-Chih Chen, Tai-Jan Chiu, Yu-Ming Wang, Ching-Nung Wu, Yao-Hsu Yang, Sheng-Dean Luo, Shao-Chun Wu

**Affiliations:** 1Department of Internal Medicine, Kaohsiung Chang Gung Memorial Hospital and Chang Gung University College of Medicine, Kaohsiung 833, Taiwan; b9902018@cgmh.org.tw; 2Department of Otolaryngology, Kaohsiung Chang Gung Memorial Hospital and Chang Gung University College of Medicine, Kaohsiung 833, Taiwan; b0002066@cgmh.org.tw (W.-S.C.); jarva@adm.cgmh.org.tw (W.-C.C.); taytay@cgmh.org.tw (C.-N.W.); 3Department of Pathology, University of Utah, Salt Lake City, UT 84112, USA; yin.wee@hsc.utah.edu; 4School of Dentistry, College of Oral Medicine, Taipei Medical University, Taipei 110, Taiwan; chingshuenwang@tmu.edu.tw; 5Department of Hematology-Oncology, Kaohsiung Chang Gung Memorial Hospital and Chang Gung University College of Medicine, Kaohsiung 833, Taiwan; kuerten@cgmh.org.tw; 6Graduate Institute of Clinical Medical Sciences, College of Medicine, Chang Gung University, Taoyuan 333, Taiwan; 7Department of Radiation Oncology and Proton & Radiation Therapy Center, Kaohsiung Chang Gung Memorial Hospital and Chang Gung University College of Medicine, Kaohsiung 833, Taiwan; scorpion@cgmh.org.tw; 8Department of Traditional Chinese Medicine, Chang Gung Memorial Hospital, Chiayi 613, Taiwan; gmailr95841012@adm.cgmh.org.tw; 9Health Information and Epidemiology Laboratory of Chang Gung Memorial Hospital, Chiayi 613, Taiwan; 10School of Traditional Chinese Medicine, College of Medicine, Chang Gung University, Taoyuan 333, Taiwan; 11Department of Anesthesiology, Kaohsiung Chang Gung Memorial Hospital and Chang Gung University College of Medicine, Kaohsiung 833, Taiwan

**Keywords:** direct oral anticoagulants, warfarin, survival, head and neck cancer, cancer-associated thromboembolism

## Abstract

**Simple Summary:**

Patients with head and neck cancers may suffer from cancer-associated thromboembolism and direct oral anticoagulants (DOACs) are a potential new therapeutic option. We aimed to determine the clinical impact of DOACs compared with traditional anticoagulants on the survival of patients with head and neck cancers. In our study, DOAC users had significantly better disease-specific survival (DSS) and higher overall survival (OS) rates than warfarin users and those who did not use any anticoagulant. Further, there were no significant differences in the occurrence rate of bleeding or ischemic events between DOAC and warfarin users. Our study suggested that DOACs can be a treatment choice or prophylaxis for tumor emboli in head and neck cancer patients and that they might be a better choice than traditional anticoagulants.

**Abstract:**

Increasing clinical evidence supports the use of direct oral anticoagulants (DOACs) as a potential new therapeutic option for patients suffering from cancer-associated thromboembolism. However, the clinical impact of DOACs compared with traditional anticoagulants on the survival of patients with head and neck cancer has not been well studied. A total of 1025 patients diagnosed as having head and neck cancer, including 92 DOAC users, 113 warfarin users, and 820 nonusers of anticoagulants, were selected from the Chang Gung Research Database between January 2001 and December 2019. The patients were matched using the propensity-score method. The survival rates were estimated among the three groups using the Kaplan–Meier method. The protective effects and side effects of the two anticoagulants were compared using the chi-square test. The death rate (18 patients, 19.57%) in patients using DOACs was significantly lower than that in patients using warfarin (68 patients, 60.18%) and those not using any anticoagulant (403 patients, 49.15%). DOAC users had significantly better disease-specific survival (DSS) than warfarin users (*p* = 0.019) and those who did not use any anticoagulant (*p* = 0.03). Further, DOAC users had significantly higher overall survival (OS) rates than warfarin users and those who did not use any anticoagulant (*p* = 0.003). Patients with oropharyngeal and laryngeal cancer and DOAC users had a significantly lower hazard ratio for survival, whereas patients with American Joint Committee on Cancer stage IV disease and those receiving multidisciplinary treatment (e.g., surgery with radiotherapy or concurrent radiochemotherapy) had a significantly higher hazard ratio for survival. Among them, patients with laryngeal cancer (HR = 0.47, 95% CI = 0.26–0.86, *p* = 0.0134) and DOAC users (HR = 0.53, 95% CI = 0.29–0.98, *p* = 0.042) had the lowest hazard ratio from DSS analysis. Similarly, patients with laryngeal cancer (HR = 0.48, 95% CI = 0.30–0.76, *p* = 0.0018) and DOAC users (HR = 0.58, 95% CI = 0.36–0.93, *p* = 0.0251) had the lowest hazard ratio from OS analysis. As for the protective effects or side effects of anticoagulants, there were no significant differences in the occurrence rate of bleeding or ischemic events between DOAC and warfarin users. In our study, DOACs were found to be better than warfarin in terms of survival in patients with head and neck cancer. As regards thromboembolism prevention and side effects, DOACs were comparable to warfarin in our patients. DOACs can be a treatment choice or prophylaxis for tumor emboli in head and neck cancer patients and they might be a better choice than traditional anticoagulants according to the results of our study.

## 1. Introduction

One in five cancer patients suffers from venous thrombosis [1]. Moreover, patients with head and neck cancer have the second highest risk of cancer-associated thromboembolism. [2]. From the viewpoint of pathophysiology, there is a strong connection between the rapid progression of cancer and venous thromboembolism [3]. Clinicians have previously noted a relationship between cancer and venous thromboembolism (VTE) [4]. Cancer-induced coagulation and inflammatory processes, along with other thrombus-inducing conditions (e.g., surgery, central venous catheter placement) can increase the risk of VTE formation in cancer patients. Furthermore, studies have revealed that cancer-induced coagulation plays an important role in cancer growth and metastasis [5]. Clinically, patients with VTE were found to have a shorter overall survival (OS) than those without thrombus formation [6]. Because of the above relationship, many studies have focused on the benefits of anticoagulants for cancer treatment and prevention [4].

An increasing amount of clinical evidence has suggested that the most broadly used anticoagulants, low-molecular-weight heparin (LMWH) and warfarin, may have anticancer effects [4,7]. Hence, anticoagulants may be used to improve survival and prevent cancer progression or metastasis [4,7]. Recently, randomized clinical trials for VTE and atrial fibrillation [8,9,10] revealed that direct oral anticoagulants (DOACs) can reduce the risk of venous and arterial thromboembolism. Likewise, Kahale et al. [11] conducted a systemic review comparing the effectiveness of DOACs, LMWH, and warfarin for the treatment of VTE in patients with cancer. However, there is some new evidence from a systemic review comparing DOACs, LMWH, and warfarin regarding treatment of venous thromboembolism in people with cancer. The study revealed that DOACs, compared to LMWH, may reduce VTE but at the same time increase the risk of bleeding. However, there is more evidence that supports the safety and efficacy of DOACs in cancer patients [12]. To date, the effect of DOACs for cancer management is unclear, and there is no solid evidence that DOACs have anticancer effects similar to those of LMWH or warfarin [13]. In addition, no study has compared the safety and efficacy of warfarin and DOACs in patients with head and neck cancer.

Previous studies have showed that warfarin has no benefit for cancer patients because of bleeding risks, multiple drug–drug interactions, and food influences [14]. DOACs are known to have less drug–drug and drug–food interactions, and previous data have revealed fewer bleeding risks with DOAC use in cancer patients [15]. However, the adverse events associated with DOAC use in patients with head and neck cancer remain unclear.

In this study, we aimed to determine the clinical impact of DOACs compared with traditional anticoagulants on the survival of patients with head and neck cancer, as well as their side effects and risks.

## 2. Materials and Methods

### 2.1. Patient Recruitment

Our study was a retrospective cohort study. The data were obtained from the Chang Gung Research Database. This study was approved by the Institutional Review Board (IRB) of the Kaohsiung and Chiayi branches of Chang Gung Memorial Hospital (reference numbers: 201801348B0C601, 201700253B0C602, and 201901691B0). Between 1 January 2001 and 31 December 2019, a total of 15,637 patients diagnosed as having head and neck cancer were identified (ICD 10: C00, C02, C03, C04, C05, C06, C09, C10, C12, C13, C14, C32). Exclusion criteria were as follows: ICD 10—C07, C08, C30, C31, C058, C059 (*n* = 841); American Joint Committee on Cancer (AJCC) stage IVc or missing data regarding cancer stage (*n* = 345); nonsquamous cell carcinoma patients (*n* = 104); patients who took both DOACs and warfarin (*n* = 26); and patients who did not receive treatment (*n* = 107). After exclusion, 14,214 patients were evaluated in our study. For patients taking oral anticoagulants, the choice of one of the two anticoagulant therapies was made according to patients’ clinical condition. Because of the higher probability of drug-drug interaction and the need of frequent monitoring, DOACs were generally preferred over warfarin for patients who need oral anticoagulant therapy. We started using DOACs at our hospital since 2010. In some specific conditions, such as in patients with impaired renal function, moderate to severe mitral valve stenosis, or mechanical prosthetic valve implantation, DOACs may be avoided or used with adjusted dosage [16,17].

### 2.2. Statistical Analysis

A two-sided Fisher’s exact test or Pearson’s chi-squared test was used to evaluate demographic and categorical data, such as sex, related comorbidities, lifestyle risk factors (alcohol or betel nut consumption), and AJCC stage of cancer. Normally distributed continuous data were analyzed using Student’s *t*-test, while non-normally distributed data were analyzed by Mann–Whitney U-test. To reduce the effect of confounding factors, we created a 1:4 propensity-score-matched study group (oral anticoagulant user vs. nonuser) by using the Greedy method with a 0.25 caliper width (NCSS Statistical Software, Kaysville, UT, USA). Sex, age, and AJCC stage of the cancer were chosen as covariates and a logistic regression model was used to calculate the propensity scores. After adjusting the effect of the confounding factors, the effects of oral anticoagulant use on the primary outcome (OS and disease-specific survival [DSS]) were evaluated by the Kaplan–Meier method. As for factors that might affect survival, a univariate analysis and a Cox proportional hazards model were used for evaluation. All statistical analyses were performed using SAS 9.4 and SPSS Statistics V25.0 for Windows (IBM Corp., Armonk, NY, USA). A *p*-value < 0.05 was considered statistically significant for each analysis.

## 3. Results

### 3.1. Demographic and Clinical Characteristics of the Study Cohort

Between January 2001 and December 2019, we identified 15,637 patients diagnosed as having head and neck cancer in the Chang Gung Research Database. A flow chart of this cohort study is shown in Figure 1. After applying exclusion criteria, 14,214 patients were identified for further analysis. Patients who had ever received oral anticoagulants were categorized into the DOAC or warfarin group, and those who had not were categorized into the nontreated group (none). We performed a 1:4 propensity-score-matching analysis between the oral-anticoagulant-treated and nontreated groups of the patients. Finally, 1025 patients were recruited for this study, of whom 92 were prescribed DOACs (i.e., 21 Apixaban, 8 Dabigatran, 11 Edoxaban, and 52 Rivaroxaban), 113 were prescribed warfarin, and 820 did not use any oral anticoagulants. The average treatment durations of these four DOACs were all longer than 6 months. For warfarin, the average treatment duration was longer than 1 year. The daily dosages of DOACs and warfarin were applied according to the clinical requirements (Appendix A).

The clinical characteristics and demographic data are summarized in Table 1. Of these 1025 patients, 4.68% were female and 95.32% were male. The mean age at the time of diagnosis was 59 years. In terms of the AJCC stage, most of the patients were diagnosed as having stage IV head and neck squamous cell carcinoma (HNSCC) (492 patients, 48%). In terms of the treatment type, 420 (40.98%) patients received surgery alone, 258 (25.17%) patients received surgery plus adjuvant radiotherapy (RT) or concurrent radiochemotherapy (CCRT), and 347 (33.85%) patients received RT or CCRT without surgery. There were no significant differences in sex, cancer staging, cancer subsites, cancer recurrence, or treatments among the three groups. At the end of the study, there were 489 (47.71%) deaths, and 302 (29.46%) patients died because of head and neck cancer. The death rate (18 patients, 19.57%) among patients using DOACs was significantly lower than that among patients using warfarin (68 patients, 60.18%) and those who did not use any anticoagulant (403 patients, 49.15%). There were also significantly lower HNSCC-related death rates among patients treated with DOACs than those treated with warfarin (11.96% and 30.97%, respectively). Compared with nonusers of anticoagulants, DOACs or warfarin users had significantly higher rates of comorbidities, such as diabetes mellitus, hypertension, atrial fibrillation, and hyperlipidemia.

### 3.2. Survival Analyses

The average survival duration of the study group was 4.73 ± 3.77 years. We performed a Kaplan–Meier survival analysis to evaluate 5-year DSS and OS in the three groups. For DSS, there was no statistically significant difference among the three groups (*p* = 0.06, Table 2). However, DOAC users had significantly better DSS than did warfarin users (*p* = 0.019) and those who did not use any anticoagulant (*p* = 0.03) (Figure 2). For OS, DOAC users had significantly higher OS rates than warfarin users and those who did not use any anticoagulant (*p* = 0.003, Figure 3 and Table 3). We also analyzed the effects of four different DOACs on the OS and DSS. No significant differences were found, which could be due to the small sample size. This part is fully described in the Appendix A.

### 3.3. Cox Regression Analyses of Independent Prognostic Factors for Survival

We chose sex, age, AJCC stage, treatment strategy, and oral anticoagulant usage for the analyses. In the univariate analysis of DSS, all factors, except for sex and age, were identified as being significant prognostic factors. In the multivariate analysis, patients with oropharyngeal and laryngeal cancer and DOAC users had significantly lower hazard ratios, whereas patients with AJCC stage IV disease and those receiving multidisciplinary treatment (e.g., surgery with RT or CCRT) had a significantly higher hazard ratio (Table 4). Among them, patients with laryngeal cancer (HR = 0.47, 95% CI = 0.26–0.86, *p* = 0.0134) and DOAC users (HR = 0.53, 95% CI = 0.29–0.98, *p* = 0.042) had the lowest hazard ratio (Table 4). For OS, the multivariate analysis revealed results similar to those of the DSS analysis. Patients with laryngeal cancer (HR = 0.48, 95% CI = 0.30–0.76, *p* = 0.0018) and DOAC users (HR = 0.58, 95% CI = 0.36–0.93, *p* = 0.0251) also had the lowest hazard ratio for OS (Table 5).

On comparing DSS and OS within each AJCC stage, we found that patients with AJCC stage IV disease had the worst DSS and OS. The data indicated that AJCC stage IV disease was a significant factor for disease prognosis. To compare treatment effects, we also separated patients into three treatment groups: surgery alone; RT, chemotherapy, or CCRT; and surgery with adjuvant RT or CCRT. Patients receiving multidisciplinary treatment had a significantly higher hazard ratio than those receiving surgery alone. From the Cox regression analysis of DSS, the hazard ratio of surgery plus RT or CCRT was 2.26 and the hazard ratio of RT or CCRT alone was 6.42 (Table 5). In the analysis of OS, the risk ratio of surgery plus RT or CCRT was 1.91, and the hazard ratio of RT or CCRT was 4.36 (Table 5). There were statistically significant differences in all of these data between the treatment groups.

### 3.4. Protective Effects and Side Effects of DOACs and Warfarin

To assess the protective effects and side effects of DOACs and warfarin, we evaluated the occurrence of bleeding events (e.g., upper gastrointestinal (GI) bleeding and intracranial hemorrhage (ICH)) and ischemic diseases (e.g., myocardial infarction (MI), ischemic stroke, deep-vein thrombosis, and pulmonary embolism). There were no significant differences in the occurrence rate of bleeding or ischemic events between DOAC and warfarin users (Table 6).

## 4. Discussion

This is the first study to evaluate the influence of DOACs and warfarin in patients with head and neck cancer. On the basis of our findings, OS and DSS rates in the DOAC group were better than those in the warfarin group and the group that did not use any anticoagulant, and the side effects of DOACs and warfarin were similar. Several systemic review [18] and randomized controlled studies [19,20] have reported a lower risk of venous thromboembolism in cancer patients using DOACs, compared to LMWH or placebo therapy. Along with these studies and our study results, we believe that DOACs can perform as a possible treatment choice or prevention measure for tumor emboli in patients with head and neck cancer. In our study, clinical pathological factors, including advanced cancer stage and treatment without surgery, were associated with OS and DSS.

Head and neck cancer is the sixth most common cancer worldwide [21], including oral, oropharyngeal, hypopharyngeal, and laryngeal cancers. The most common cell type is a squamous cell carcinoma. Different cancer subsites are associated with different OS rates. Patients with laryngeal cancer had the highest survival rate in our study, which is also consistent with the results of previous studies [22]. Smoking, alcohol use, and betel nut chewing are major risk factors for head and neck cancers [23]. Other risk factors include human papilloma virus (HPV) infection, diet, physical training, and lymph node ratio [24]. A combination of surgery and RT plus chemotherapy is a negative prognostic factor for oral cancer [25]. In our study, the negative prognostic factors of OS and DSS in head and neck cancer were a late cancer stage and nonsurgical treatment, which were compatible with the findings of previous studies [22].

Compared with warfarin, DOACs have been used in many studies. DOACs are a group of anticoagulants, including dabigatran (Pradaxa^®^, approved by the FDA in 2010), rivaroxaban (Xarelto^®^, approved by the FDA in 2011), apixaban (Eliquis^®^, approved by the FDA in 2012), and edoxaban (Savaysa^®^, approved by the FDA in 2015). Dabigatran is a direct inhibitor of coagulase, whereas rivaroxaban, apixaban, and edoxaban are factor Xa inhibitors. Compared with warfarin, which is a Vitamin K inhibitor, the benefits of DOACs include a rapid onset, a short half-life, fewer drug–drug interactions, and predictable pharmacokinetic models. Clinicians do not need to monitor the international normalized ratio (prothrombin time) regularly to ensure the efficacy of the anticoagulants. For nonvalvular atrial fibrillation, DOACs have a similar effect to warfarin but are safer than warfarin [26]. In addition, using DOACs is not associated with an increased risk of major bleeding or mortality when treating VTE. In 2019, AHA/ACC/HRS renewed the treatment guidelines for atrial fibrillation and strongly recommended that patients with atrial fibrillation replace warfarin with DOAC [8]. This recommendation was compatible with the treatment of the patients in this study, and the number of patients who used DOACs was higher than those using warfarin.

In our study, warfarin and DOACs were found to be similarly effective in protecting against VTE, coronary artery disease, and ischemic stroke. In terms of the side effects, including major bleeding such as GI bleeding or ICH, there was no statistically significant difference between warfarin and DOACs.

The condition may become different when DOACs are used to replace warfarin for the prevention and treatment of venous thromboembolism. Cancer patients are at a high risk of mortality because of thrombus formation. A previous study showed the relationship between cancer and high thrombosis status; there is four to seven times higher risk of VTE in cancer patients [27]. It is difficult to decide when to start VTE treatment and which type of anticoagulant should be used. A previous study indicated that cancer patients, particularly late cancer stage or very-early-stage patients, may benefit from LMWH and warfarin [4,28]. However, evidence on the safety of DOAC use in patients with cancer is limited. Researchers have questioned the safety and efficacy of DOACs in cancer patients [29,30]. Recently, new evidence has supported the safety and efficacy of DOACs [12] and proved that DOACs are useful for the prevention and treatment of cancer-related VTE [19,20]. For atrial fibrillation patients and cancer patients, the risk of bleeding or stroke is similar or lower when DOAC users are compared with warfarin users [31]. Furthermore, the American Society of Clinical Oncology (ASCO) changed their previous recommendations and suggested that clinical physicians offer apixaban or rivaroxaban for VTE prevention in high-risk patients. Rivaroxaban and edoxaban are new VTE choices as well [32]

Most clinical studies indicate that the risk of thromboembolism formation in patients with head and neck cancer is very low, and the risk is the lowest among all types of cancers [33]. However, in one study, which focused on the relationship between VTE and hand and neck cancer in patients who underwent major surgery, a VTE incidence of up to 26.3% was found [34]. In addition, head and neck cancer patients have biological factors supporting high risk of thromboembolism, including strong presentation of procoagulant proteins, better thromboembolism formation mechanism, fibrinolysis mechanism, and procoagulant factor secretion [35]. This is paradoxical. One possible explanation is that those previous studies might have been affected by some factors. For example, mixed cell types of head and neck cancers, including squamous cell carcinoma, adenocarcinoma, and nasopharyngeal cancer, were all included in the studies. Different cell types of cancer result in different risks of VTE formation. Another explanation is that head and neck cancer and VTE have similar risk factors, such as smoking, old age, and long-term treatment of head and neck cancer. Because of the lack of clinical evidence, there is no solid evidence for clinical recommendations regarding VTE in patients with head and neck cancer. In our study, the major side-effects (e.g., ICH or GI bleeding) were not statistically different between warfarin and DOAC users. In these two groups of anticoagulant users, the effects of prevention of VTE or major illnesses such as cardiovascular disease (including MI and ischemic stroke) are similar as well. On the basis of our study results, DOACs may be an option for VTE treatment or prevention.

The potential effect of DOACs on cancer inhibition requires further investigation. In our study, the OS and DSS of the DOAC group were the highest. Stopping the process of angiogenesis may restrict the growth of malignant tumors. The interaction of factors for angiogenesis occurs through protease-activated receptors (PARs) on the surface of tumor cells. By degeneration or activation of PARs, antithrombins (e.g., factor Xa and anti-coagulase) help the process of angiogenesis, inflammation, and fibrosis and then stimulate tumor progression. On the basis of this theory, inhibition of factor Xa or anti-coagulase by DOACs may help in cancer inhibition [13]. Our DSS analysis results were compatible with the theory that factor Xa inhibitors may inhibit cancer progression. However, previous data from animal models have showed no similar results, indicating that the effect of DOACs on tumor progression and metastasis depended on the time of DOAC use and the model of tumor cells [36]. This is different from the results of our study. Further studies are warranted to prove the theory of cancer inhibition by Xa inhibitors.

A previous study showed that low-dose warfarin-mediated Axl inhibition was effective as an anticancer agent in vitro [37]. The precise mechanisms by which DOACs exert their anticancer effects are yet to be elucidated. It is possible that DOACs and warfarin may appear to have anticancer effects that are exerted through different mechanisms [38]. Such differences may lead to different survival outcomes in patients with head and neck cancers. The use of DOACs in patients with head and neck cancers resulted in better overall survival compared with warfarin, which may also suggest that the beneficial effect of DOACs may be related to their anticoagulant function in preventing VTE. Similar to our results, a previous study showed that cancer patients with atrial fibrillation in the warfarin group had a higher mortality rate than those in the DOAC group [39]. The study supported the notion that the use of warfarin was associated with an increased risk of thromboembolic events compared with the use of DOACs. Although the risk of side effects such as bleeding was not significantly different between the warfarin and the DOAC groups in our study, some studies have revealed that DOAC users had lower mortality rates and less bleeding events when compared with warfarin users [11]. These findings suggest that warfarin may be linked to more severe, potentially fatal, bleeding than DOACs. Therefore, it appears that this factor may also influence the survival outcomes in cancer patients. There were some limitations to our study. First, this is a retrospective study. The outcome could be affected by the choice of anticoagulant therapy itself, and there are still possible factors that were not identified in our study. Second, in the OS and DSS analyses, we could not perform a direct analysis for head and neck cancer because of the small sample size. Third, we did not have HPV data, which is a prognostic factor for head and neck cancer. In addition, we started using DOACs at our hospital since 2010, which is one of the reasons why we had a relatively small sample size and short follow-up period. Finally, we had to give up some data because of incomplete data collection. This is one of the disadvantages of big data studies.

## 5. Conclusions

This is a retrospective cohort study of the clinical impact of DOACs compared with traditional anticoagulants on the survival of patients with head and neck cancer. The results showed that OS and DSS were higher in the DOAC group than in the warfarin group and the group that did not use any anticoagulant. The side effects of DOACs and warfarin were similar. Further studies are warranted to prove the theory of cancer inhibition by factor Xa inhibitors.

## Figures and Tables

**Figure 1 cancers-14-00703-f001:**
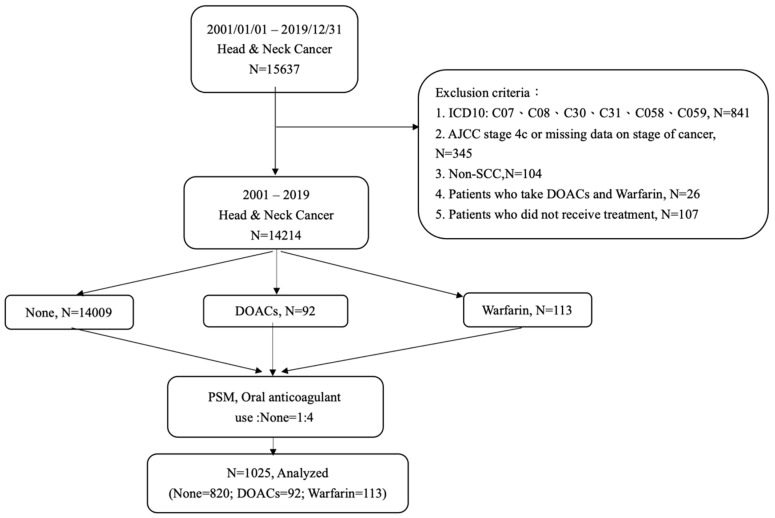
Flowchart of head and neck cancer patient inclusion and exclusion in the study cohort. (AJCC stage—American Joint Committee on Cancer stage; SCC—squamous cell carcinoma; DOACs—direct oral anticoagulants; PSM—propensity-score-matched study).

**Figure 2 cancers-14-00703-f002:**
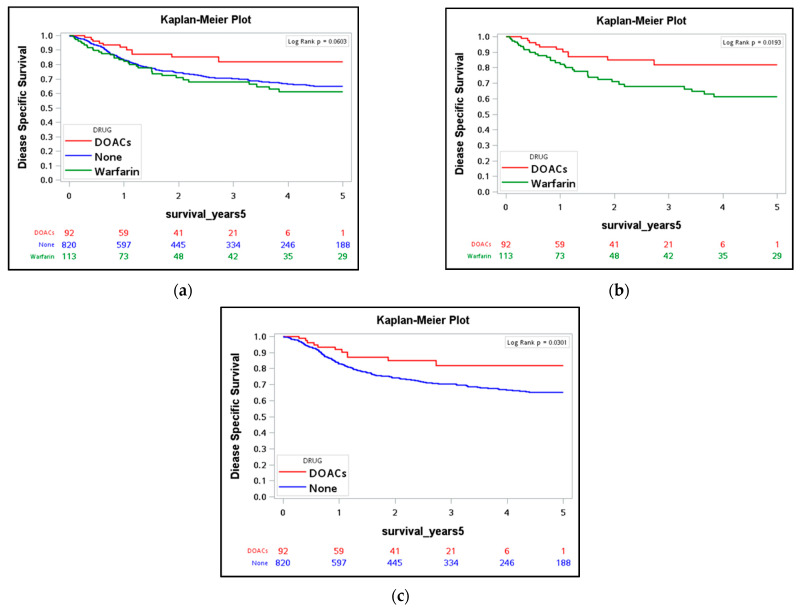
Kaplan–Meier survival curve of disease specific survival between **(a**) DOACs users (*n* = 92), warfarin users (*n* = 113) and non-users (*n* = 820); (**b**) DOACs users and warfarin users; (**c**) DOACs users and non-users.

**Figure 3 cancers-14-00703-f003:**
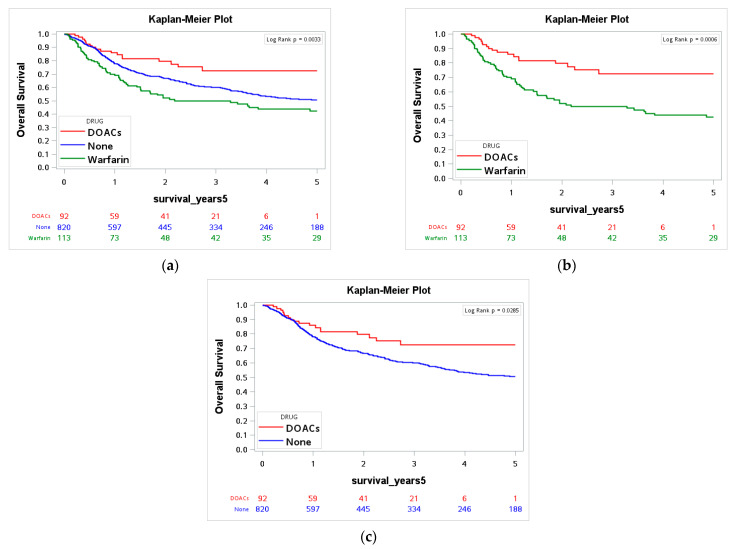
Kaplan–Meier survival curve of overall survival between (**a**) DOACs users (*n* = 92), warfarin users (*n* = 113) and non-users (*n* = 820); (**b**) DOACs users and warfarin users; (**c**) DOACs users and non-users.

**Table 1 cancers-14-00703-t001:** Demographic and clinical characteristics of the study cohort (*n* = 1025).

Variables		N (%)	Oral Anticoagulants			*p*-Value	
			None	DOACs	Warfarin		
Sex	Femal	48 (4.68%)	36 (4.39%)	6 (6.52%)	6 (5.31%)	0.6207	
	Male	977 (95.32%)	784 (95.61%)	86 (93.48%)	107 (94.69%)		
Age at diagnosis (Mean ± SD)	Years	59.4 ± 11.4	59.3 ± 11.5	62.1 ± 10.7	57.9 ± 11.2	0.0174 *	
AJCC stage	I	232(22.63%)	187 (22.8%)	25 (27.17%)	20 (17.7%)	0.2634 ^†^	0.1507
	II	148 (14.44%)	114 (13.9%)	20 (21.74%)	14 (12.39%)	0.1031 ^†^	
	III	153 (14.93%)	122 (14.88%)	10 (10.87%)	21 (18.58%)	0.3036 ^†^	
	IV (excluding IVc)	492 (48%)	397 (48.41%)	37 (40.22%)	58 (51.33%)	0.2478 ^†^	
AJCC stage	I and II	380 (37.07%)	301 (36.71%)	45 (48.91%)	34 (30.09%)	0.0189 *	
	III and IV	645 (62.93%)	519 (63.29%)	47 (51.09%)	79 (69.91%)		
Cancer subsite	Oral cavity	745 (72.68%)	604 (73.66%)	62 (67.39%)	79 (69.91%)	0.4013	
	Oropharynx	110 (10.73%)	87 (10.61%)	13 (14.13%)	10 (8.85%)		
	Hypopharynx	100 (9.76%)	74 (9.02%)	9 (9.78%)	17 (15.04%)		
	Larynx	70 (6.83%)	55 (6.71%)	8 (8.7%)	7 (6.19%)		
Cancer Recurrence	No	828 (80.78%)	662 (80.73%)	79 (85.87%)	87 (76.99%)	0.2751	
	Yes	197 (19.22%)	158 (19.27%)	13 (14.13%)	26 (23.01%)		
Death	No	536 (52.29%)	417 (50.85%)	74 (80.43%)	45 (39.82%)	<0.0001 *	
	Yes	489 (47.71%)	403 (49.15%)	18 (19.57%)	68 (60.18%)		
Cause of death	Alive	536 (52.29%)	417 (50.85%)	74 (80.43%)	45 (39.82%)	<0.0001 *	
	Death due to HNSCC	302 (29.46%)	256 (31.22%)	11 (11.96%)	35 (30.97%)		
	Other cause of death	187 (18.24%)	147 (17.93%)	7 (7.61%)	33 (29.2%)		
Treatments	Surgery	420 (40.98%)	346 (42.2%)	38 (41.3%)	36 (31.86%)	0.2144	
	RT, CT, CCRT	347 (33.85%)	274 (33.41%)	33 (35.87%)	40 (35.4%)		
	Surgery + RT or CCRT	258 (25.17%)	200 (24.39%)	21 (22.83%)	37 (32.74%)		
Smoking (*n* = 902)	No	260 (28.82%)	198 (27.69%)	21 (23.6%)	41 (41.84%)	0.0077 *	
	Yes	642 (71.18%)	517 (72.31%)	68 (76.4%)	57 (58.16%)		
Betel nuts consumption (*n* = 916)	No	435 (47.49%)	341 (47.1%)	35 (38.04%)	59 (59%)	0.0132 *	
	Yes	481 (52.51%)	383 (52.9%)	57 (61.96%)	41 (41%)		
Alcoholic beverages (*n* = 916)	No	405 (44.21%)	316 (43.65%)	35 (38.04%)	54 (54%)	0.0673	
	Yes	511 (55.79%)	408 (56.35%)	57 (61.96%)	46 (46%)		
DM	No	795 (77.56%)	654 (79.76%)	64 (69.57%)	77 (68.14%)	0.0033 *	
	Yes	230 (22.44%)	166 (20.24%)	28 (30.43%)	36 (31.86%)		
Hypertension	No	700 (68.29%)	610 (74.39%)	39 (42.39%)	51 (45.13%)	<0.0001 *	
	Yes	325 (31.71%)	210 (25.61%)	53 (57.61%)	62 (54.87%)		
Atrial fibrillation (flutter)	No	913 (89.07%)	793 (96.71%)	35 (38.04%)	85 (75.22%)	<0.0001 *	
	Yes	112 (10.93%)	27 (3.29%)	57 (61.96%)	28 (24.78%)		
Hyperlipidemia	No	846 (82.54%)	705 (85.98%)	64 (69.57%)	77 (68.14%)	<0.0001 *	
	Yes	179 (17.46%)	115 (14.02%)	28 (30.43%)	36 (31.86%)		

Abbreviations: DOACs—direct oral anticoagulants; SD—standard deviation; AJCC stage—American Joint Committee on Cancer stage; RT—radiotherapy; CT—chemotherapy; CCRT—concurrent radio-chemotherapy; DM—diabetes mellitus; * *p* ≤ 0.05; ^†^ respective *p*-value in each AJCC group.

**Table 2 cancers-14-00703-t002:** Disease-specific survival of DOAC users, warfarin users, and nonusers (*n* = 1025).

Variables	Cohort N = 1025	Survival Rate (%) Years					*p*-Value
		1	2	3	4	5	
None	820 (80.00%)	83.0	74.2	70.3	66.5	65.0	*p* = 0.0603
DOAC use	92 (8.98%)	91.9	85.2	82.0	82.0	82.0	
Warfarin use	113 (11.02%)	82.3	71.0	67.9	61.3	61.3	

Abbreviations: DOACs—direct oral anticoagulants.

**Table 3 cancers-14-00703-t003:** Overall survival between DOACs users, warfarin users and non-users. (*n* = 1025).

Variables	Cohort N = 1025	Survival Rate (%) Years					*p*-Value
		1	2	3	4	5	
None	820 (80.00%)	77.9	66.7	60.0	53.4	50.6	*p* = 0.0033 *
DOACs use	92 (8.98%)	85.9	79.6	72.6	72.6	72.6	
Warfarin use	113 (11.02%)	68.9	52.1	49.8	43.8	42.4	

Abbreviations: DOACs—direct oral anticoagulants; * *p* ≤ 0.05.

**Table 4 cancers-14-00703-t004:** Univariate and multivariate Cox regression model of prognostic factor for disease specific survival in study cohort (*n* = 1025).

Variables	Comparison	N/Mean ± SD	Hazard Ratio (95% CI)			
			Univariate	*p*	Multivariate	*p*
Sex	Female	48 (4.68%)	1		1	
	Male	977 (95.32%)	1.08 (0.60–1.92)	0.8024	0.95 (0.52–1.73)	0.8584
Age	Years	59 (52–67)	0.99 (0.98–1.01)	0.2948	1 (0.99–1.01)	0.7964
Cancer subsite	Oral cavity	745 (72.68%)	1	<0.0001 *	1	0.0092 *
	Oropharynx	110 (10.73%)	2.10 (1.52–2.91)	<0.0001 *	0.75 (0.52–1.1)	0.1457
	Hypopharynx	100 (9.76%)	1.70 (1.19–2.42)	0.0034 *	0.56 (0.38–0.84)	0.0049 *
	Larynx	70 (6.83%)	0.80 (0.46–1.37)	0.4138	0.47 (0.26–0.86)	0.0134 *
AJCC stage	I	232 (22.63%)	1	<0.0001 *	1	0.0105 *
	II	148 (14.44%)	1.76 (1.02–3.03)	0.0420 *	1.35 (0.77–2.35)	0.2921
	III	153 (14.93%)	1.91 (1.13–3.22)	0.0162 *	1.09 (0.63–1.9)	0.7620
	IV (IVA and IVB)	492 (48%)	4.56 (3.02–6.87)	<0.0001 *	1.88 (1.15–3.07)	0.0113 *
Treatment	Surgery	420 (40.98%)	1	<0.0001 *	1	<0.0001 *
	Surgery + RT and CCRT	347 (33.85%)	2.99 (2.11–4.22)	<0.0001 *	2.26 (1.51–3.38)	<0.0001 *
	RT, CT, and CCRT	258 (25.17%)	6.48 (4.63–9.06)	<0.0001 *	6.42 (4.09–10.09)	<0.0001 *
Oral anticoagulants	None	820 (80.00%)	1	0.0666	1	0.1168
	DOACs	92 (8.98%)	0.52 (0.28–0.95)	0.0331 *	0.53 (0.29–0.98)	0.042 *
	Warfarin	113 (11.02%)	1.16 (0.80–1.66)	0.4389	1.05 (0.72–1.51)	0.807

Abbreviations: SD—standard deviation; DOACs—direct oral anticoagulants; AJCC stage—American Joint Committee on Cancer stage; RT—radiotherapy; CT—chemotherapy; CCRT—concurrent radiochemotherapy; * *p* ≤ 0.05.

**Table 5 cancers-14-00703-t005:** Univariate and multivariate Cox regression model of prognostic factor for overall survival in study cohort (*n* = 1025).

Variables	Comparison	N/Mean ± SD	Hazard Ratio (95% CI)			
			Univariate	*p*	Multivariate	*p*
Sex	Female	48 (4.68%)	1		1	
	Male	977 (95.32%)	1.10 (0.69–1.77)	0.6881	1.21 (0.74–1.97)	0.4424
Age	Years	59 (52–67)	1.01 (1.00–1.02)	0.0452 *	1.02 (1.01–1.03)	0.0019 *
Cancer subsite	Oral cavity	745 (72.68%)	1	<0.0001 *	1	0.0039 *
	Oropharynx	110 (10.73%)	1.88 (1.43–2.47)	<0.0001 *	0.82 (0.59–1.13)	0.2298
	Hypopharynx	100 (9.76%)	1.62 (1.21–2.16)	0.0013 *	0.65 (0.47–0.91)	0.0109 *
	Larynx	70 (6.83%)	0.88 (0.58–1..33)	0.5428	0.48 (0.30–0.76)	0.0018 *
AJCC stage	I	232 (22.63%)	1	<0.0001 *	1	0.0307 *
	II	148 (14.44%)	1.43 (0.96–2.12)	0.0794	1.13 (0.75–1.69)	0.5641
	III	153 (14.93%)	1.72 (1.19–2.49)	0.0042 *	1.13 (0.76–1.67)	0.5535
	IV (IVA and IVB)	492 (48%)	3.01 (2.25–4.02)	<0.0001 *	1.56 (1.09–2.22)	0.0144 *
Treatment	Surgery	420 (40.98%)	1	<0.0001 *	1	<0.0001 *
	Surgery + RT and CCRT	347 (33.85%)	2.23 (1.73–2.88)	<0.0001 *	1.91 (1.41–2.58)	<0.0001
	RT, CT, and CCRT	258 (25.17%)	4.38 (3.41–5.63)	<0.0001 *	4.36 (3.08–6.17)	<0.0001
Oral anticoagulants	None	820 (80.00%)	1	0.0038 *	1	0.0101 *
	DOACs	92 (8.98%)	0.59 (0.37–0.94)	0.0281 *	0.58 (0.36–0.93)	0.0251 *
	Warfarin	113 (11.02%)	1.39 (1.05–1.83)	0.0204 *	1.30 (0.99–1.72)	0.0642

Abbreviations: SD—standard deviation; DOACs—direct oral anticoagulants; AJCC stage—American Joint Committee on Cancer stage; RT—radiotherapy; CT—chemotherapy; CCRT—concurrent radiochemotherapy; * *p* ≤ 0.05.

**Table 6 cancers-14-00703-t006:** Relationship between DOAC or Warfarin use and bleeding (or ischemic) events.

Event		Oral Anticoagulant		*p*-Value
		DOACs	Warfarin	
UGI bleeding (OPD)	No	57 (61.96%)	79 (69.91%)	0.2306
	Yes	35 (38.04%)	34 (30.09%)	
UGI bleeding (Admission)	No	87 (94.57%)	109 (96.46%)	0.5189
	Yes	5 (5.43%)	4 (3.54%)	
MI	No	90 (97.83%)	110 (97.35%)	1.0000
	Yes	2 (2.17%)	3 (2.65%)	
ICH	No	89 (96.74%)	111 (98.23%)	0.6588
	Yes	3 (3.26%)	2 (1.77%)	
CVA	No	75 (81.52%)	95 (84.07%)	0.6295
	Yes	17 (18.48%)	18 (15.93%)	
DVT	No	84 (91.3%)	93 (82.3%)	0.0619
	Yes	8 (8.7%)	20 (17.7%)	
PE	No	88 (95.65%)	107 (94.69%)	1.0000
	Yes	4 (4.35%)	6 (5.31%)	

Abbreviations: UGI, upper gastrointestinal; MI, myocardial infarction; ICH, intracerebral hemorrhage; CVA, cerebrovascular accident; DVT, deep-vein thrombosis; PE, pulmonary embolism.

## Data Availability

Restrictions apply to the availability of these data. Data was obtained from Chang Gung Research Database and are available with the permission of Institutional Review Board (IRB) of the Kaohsiung and Chiayi branches of Chang Gung Memorial Hospital.

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
