# Peer review of "Direct Oral Anticoagulants Are Associated with Superior Survival Outcomes than Warfarin in Patients with Head and Neck Cancers"

_cancers, 2022, doi:10.3390/cancers14030703_

Round 1
Reviewer 1 Report
This is a retrospective cohort study on a group of selected head and neck cancer patients between 2001 and 2019 in one major cancer center in Taiwan. The authors analyzed and compared the survival rate among users of three different categories of anticoagulants: DOACS, warfarin, and none and found a significant survival benefit on DOACS users than warfarin and none.
There are some major concerns:
- there is no clear description on the selection criteria of three different anticoagulant group, including the duration, frequency, etc of the usage of three different anticoagulants.
- there is no explanation on the possible mechanism that DOACS is better than warfarin in users' survival.
There are some minor concerns:
- Please check English spelling, grammar and flow.
Reviewer 2 Report
Introduction
In order to improve the literature reported on this subject, it would be interesting to discuss this remarkable systematic review. cite doi: 10.1002/14651858.CD006650.pub5.
Methods
- the strobe guidelines were performed?
- in the figure 1 all the abbreviations need to be added in the captation
Results
- Table I add also the SD for age and other values
- AJCC stage, calculate the p value for each class and add a symbol
Discussion
- line 242, before drawing conclusions it would be better to insert the data reported in the literature as prospective or randomized studies, meta-analyzes, and then report its data. cite doi:10.1182/blood.2020005819.
- line 256, although head and neck squamous carcinoma is mainly related to viral phenotype or smoking alcohol other important indices in the patient's prognosis are locoregional or distant metastases, with interesting parameters such as lymph node ratio. cite doi:10.1016/j.anl.2021.05.007.
- line 264, it should be interesting to describe different survivals according to the different DOAC available
Round 2
Reviewer 1 Report
Accept as it is.
Author Response
We thank the reviewer again for valuable comments and the acceptance of our manuscript.